# Motor Control Evaluation as a Significant Component in Upper Limb Function Assessment in Female Breast Cancer Patients after Mastectomy

**DOI:** 10.3390/healthcare9080973

**Published:** 2021-07-31

**Authors:** Maciej Śliwiński, Piotr Wąż, Wojciech Zaręba, Rita Hansdorfer-Korzon

**Affiliations:** 1Department of Physiotherapy, Faculty of Health Sciences, Medical University of Gdansk, Debinki 7, 80-211 Gdansk, Poland; wojciech.zareba@gumed.edu.pl (W.Z.); rita.hansdorfer-korzon@gumed.edu.pl (R.H.-K.); 2Department of Nuclear Medicine and Radiology Informatics, Faculty of Health Sciences, Medical University of Gdansk, Debinki 7, 80-211 Gdansk, Poland; piotr.waz@gumed.edu.pl

**Keywords:** motor control, carcinoma, breast cancer, functional assessment, shoulder complex

## Abstract

Breast cancer is the most prevalent malignancy among women. Conservative and operative treatment methods are associated with a risk of side effects pertaining to the shoulder complex. The surgery complications including chronic pain, upper limb and chest lymphedema, range of motion limitations, and motor control deficiencies may lead to upper limb function impairment and affect the quality of life negatively. Twenty-three women were examined in the tested group and twenty-two women in the control group. The motor control was assessed with dissociation tests as defined by Comerford and Mottram. In order to assess patient-perceived upper limb disability, the Disabilities of the Arm, Shoulder and Hand (DASH) questionnaire was used. The test of glenohumeral (GH) abduction control in frontal plane and in scapular plane and medial rotation control outcomes were found to be statistically significant. It pertains to both: Comparison between groups and analysis within the tested group—body sides comparison. The DASH questionnaire results analysis indicates that there was a higher degree of subjectively perceived disability of upper limb in the tested group. Surgical interventions in the breast cancer treatment and other medical procedures affect the level of motor control and perceived disability of upper limb negatively in this group of patients. Movement faults are statistically more prevalent in the tested group. Movement faults are more prevalent on the operated side in the tested group.

## 1. Introduction

Breast cancer is the most prevalent malignancy among women [1]. However, due to significant advances in both operative and nonoperative treatment methods, the survival rate has improved greatly [2]. Both conservative and operative treatment methods are associated with a risk of side effects pertaining to the shoulder complex [3]. The severity and spread of cancer disease forces the usage of specific treatment procedures to preserve patients’ health and prolong life. The resulting side effects are often inevitable. The entire multidisciplinary team is responsible for combating the side effects in order to improve the quality of life as much as possible [4]. 

The advances in surgical techniques and minimizing their invasiveness result in less severe tissue damage in the operated areas. The complications including chronic pain, upper limb and chest lymphedema, range of motion limitations, and motor control deficiencies may lead to upper limb function impairment and affect the quality of life negatively [5,6,7]. It is common that the patients experience fear of upper limb use in daily life even many years after the treatment [8]. Clinical experience suggests that a number of upper limb function impairments may appear after a prolonged period of time [6,9]. To maintain the functional level of the upper limb, multiple elements of the movement system need to perform accordingly [10]. 

Adequate motor control is necessary for the optimization of upper limb movement patterns, which helps the load spread evenly among multiple movement system components [10]. The presence of uncontrolled movement may cause tissue overload that in turn can lead to pain, damage, and various musculoskeletal conditions. Motor dysfunction is described as a multi-faceted impairment pertaining to various aspects of the movement system. It follows that the physical therapist is required to examine not only the ranges of motion but also assess the movement patterns, identify uncontrolled movement, and determine its clinical relevance to the patient’s symptoms. Motor control assessment is essential for functional diagnosis, which provides the base for the rehabilitation program [10].

Patients that have finished the cancer treatment successfully expect a fully active life, which presents significant challenges in modern physical therapy. These challenges become more apparent when it comes to women after breast cancer surgery. With the introduction of new methods of physiotherapy referring to movement assessment and optimization together with the advantage of technological advances such as the ability to record the physical examination using cameras, therapy outcomes are now greatly improved in comparison to previous years. Still, unfortunately, the greatest difficulty in physiotherapy treatment optimization comes from the lack of assessment standards in this group of patients. There is no clinical research available focusing on the assessment of uncontrolled movement in the shoulder girdle area using dissociation tests according to the methodology described by Mottram and Comerford [10]. As a result, we could not juxtapose our outcomes with the existing literature since there is a lack of appropriate scientific papers on the subject. The lack of standardized evaluation procedures in the literature, as well as pain related function impairment and patient perceived upper limb dysfunction, stimulated us to launch a study aimed primarily at highlighting the utility and need for routine use of motor control tests. The overriding aim of this study is to identify specific uncontrolled movements in the area of the shoulder complex in a group of patients after surgical treatment of breast cancer. At the same time, it is important to indicate that such disorders occur and require motor re-education on the part of physiotherapists to make the rehabilitation process more effective.

## 2. Materials and Methods

### 2.1. Description of the Groups

The independent Bioethics Committee for Scientific Research at Medical University of Gdańsk granted a permission for the research to be conducted (permission number NKB BN/476/2018). All of the participants have expressed informed consent to take part in the research. The following study eligibility criteria for inclusion and exclusion were adopted. Inclusion criteria include informed consent to participate in the study; age of majority; and past surgical treatment for breast cancer treatment in the form of mastectomy or sparing treatment. Exclusion criteria include lack of informed consent for participation in the study; for the study group—history of reconstructive procedure done with previous breast cancer surgery; less than 2 years of recovery time after surgery; injuries in the form of fractures; dislocations in the shoulder girdle; and shoulder pain before breast cancer surgery.

#### 2.1.1. Tested Group

The participants consisted of 23 women—67 years old on average. All of the subjects had been treated for breast cancer operatively. In addition, they all had pain in the shoulder girdle and/or upper extremity on the operated side in the last 3 months. None of the subjects had pain before the surgery. Moreover, all of the women in the study group underwent axillary lymph node dissection.

#### 2.1.2. Control Group

There were 22 women in the control group—54 years old on average. The subjects had never suffered from breast cancer nor had they been operated for it. The study was conducted between early 2019 and January 2020.

### 2.2. Methods

#### 2.2.1. History Taking

History taking was the first part of the patients’ examination. It included basic questions, previous diagnosis, and treatment history of patients in the tested group. Not all of the answers were used in this paper, but they were necessary for the exclusion/inclusion criteria. The history taking card is included in the Appendix A.

#### 2.2.2. Report Upper Limb Disability Questionnaire—DASH Questionnaire

In order to assess patient-perceived upper limb disability, the Disabilities of the Arm, Shoulder and Hand (DASH) questionnaire was used, which is available through https://dash.iwh.on.ca, accessed on 6 December 2020 [11]. The participants were given 30 questions and the DASH score was calculated according to the instructions. DASH is a useful diagnostic tool in patients after breast cancer surgery [12,13].

#### 2.2.3. Physical Examination

##### Physical Examination Recording Methods

All of the physical examination elements were video-recorded using one or two cameras depending on the stage of examination. Digital material was used for later analysis and passive and active movement measurements. Additionally, the records were used for motor control assessment. The main recording device was Sony RX100 IV digital camera. The accessory device used for humeral rotation evaluation in the abduction test was Xiaomi YI lite. The cameras’ placement was standardized to optimize the reliability of the examination.

##### Shoulder Girdle Motor Control Assessment

The motor control was assessed with dissociation tests as defined by Comerford and Mottram. An adequate motor control in the shoulder girdle is present when the subject is able to hold a scapula in neutral position, while simultaneously moving through the range of glenohumeral joint. The tests are used to assess neuromuscular control around the shoulder girdle during glenohumeral motion [10]. The test comprises of two-element qualifications. Both requirements can either be passed or failed. The first assessed element is related to the angular range of motion, while the other one to the quality of movement. Each motor control test has its own range of motion benchmark. The benchmark is defined as the minimal range of motion required to attain without neutral position loss in the specified area. Therefore, the benchmark is reached when the subject is able to move through the entire predefined range without alignment disturbance—i.e., change of scapula position in this case. If the quantity outcome is favorable the examiner proceeds to assess the movement quality. This criterion is met when none of the compensation patterns is present during the movement. The compensation patterns are defined by the dissociation test authors. An example of the test is shown in Figure 1.

Shoulder girdle motor control tests used for this paper included: Arm abduction test in frontal plane and in scapular plane, arm extension test, kinetic lateral rotation test, and kinetic medial rotation test of the shoulder. Tests were performed on both sides of the body. The benchmarks used in the paper were the same as defined by the test authors. For abduction tests, the required range was 90 degrees of independent abduction. For medial rotation, the test required range was 60 degrees, while for lateral rotation 45 degrees. The benchmark for the extension test was 15 degrees. Before active range testing, passive mobility was examined in a given direction. In case there was not enough passive mobility for the test’s benchmark, the active part of the test was skipped. Only when the benchmark was met in the passive test, the active motor control was evaluated. Before the start of each test, the examiner instructed the subject and assisted in scapula positioning. After this preparation, the subject performed the task in a seated position [10]. The test was video-recorded for further analysis. The video material was analyzed and the movements were evaluated by a physical therapist specializing in motor control assessment. The possible outcomes were: “vv”—both requirements—quantitative and qualitative were met; “vx”—quantitative criterion was met, yet the quality of movement was poor; finally “xx”—when the subject failed in both aspects [10]. The “xx” outcome means that the dissociation is lacking. Such an outcome will also be called “movement fault” in the following paragraphs. This outcome denotes a movement impairment.

The result VV denotes the ability to actively maintain the neutral position of the scapula during glenohumeral abduction of 90 degrees. The XX score means that there is a lack of dissociation during the shoulder abduction movement—the scapula movement occurs before reaching the abduction benchmark range.

#### 2.2.4. Statistical Analysis

The final results were generated using the R statistics language [14]. For quantitative variables, basic statistics, i.e., mean, median, standard deviation, and the minimum and the maximum values, were calculated. Using the Shapiro-Wilk test, it was determined whether the values of the analyzed variables came from a population with a normal distribution. The differences between the groups of quantitative variables were tested using the student’s t-test or Wilcoxon rank sum test. The type of the above-mentioned tests (and additional options) were selected depending on the *p*-value of the Shapiro-Wilk test and on the homogeneity of the variance test. The Fisher’s exact test for count data was used to examine the independence of the qualitative variables collected in Table 1 and Table 2 presenting the count data of individual groups. In addition, the results are presented in the form of a mosaic diagram (Appendix A), which are included in the Appendix A. In the mosaic chart, each cell of the contingency table is represented by a rectangle. The rectangles rounded by solid lines indicate the numbers that are larger than expected. The rectangles rounded by dot dashed lines show the cases for which the numbers are smaller than expected. As can be seen in this figure, the height of the rectangle is proportional to Pearson’s residual and a width is proportional to the root of the expected value. Therefore, the area is equal to the difference between the observed and expected frequencies. The assumed significance level is α = 0.05.

## 3. Results

### 3.1. Motor Control in the Shoulder Girdle

The arm abduction control test in frontal plane and in scapular plane, as well as the internal rotation control test outcomes were found to be statistically significant. It pertains to both comparison between groups and analysis within the tested group—body sides comparison. The arm extension test and lateral rotation test outcomes were not found to be significant. The different number of analyzed cases in individual tests results from the failure to meet the passive range of motion benchmark in the pretest assessment.

Possible results of tests include: “vv”—both requirements—quantitative and qualitative were met; “vx”—quantitative criterion was met, yet the quality of movement was poor; finally “xx”—when the subject failed in both aspects. The “xx” outcome means that the dissociation is lacking.

### 3.2. DASH Questionnaire

The Disabilities of the Arm, Shoulder and Hand (DASH) questionnaire was used among 45 patients (23 in the tested group and 22 in the control group). The mean values of the test results were 37.07 in the tested group and 0.68 in the control group.

## 4. Discussion

Proper integration on all the levels of the movement system is required for appropriate function in the shoulder girdle and the upper limb [10]. Among the patients with a history of breast cancer surgery the function of the upper limb may become impaired [15]. The surgery itself and radiotherapy both lead to scar formation which in turn limits the range of motion and impairs shoulder function significantly. Diminished fascial sliding ability together with scar formation around brachial plexus nerves may lead to the development of nerve entrapment syndromes, which can generate pain and neurological symptoms such as sensory deficits, muscle weakness, neurogenic and neuropathic pain [16,17,18]. Another factor contributing to the dysfunction of the shoulder among breast cancer patients is lymphedema that can affect the upper limb, shoulder girdle, and chest [19,20,21]. The aforementioned phenomena have been scientifically investigated for many years. Scientific reports specify various consequences of surgical treatment in breast cancer patients including deterioration of joint mobility; impairment of myofascial flexibility; loss of myofascial sliding ability; and chronic pain [22,23]. Although surgical treatment methods are constantly improved and their invasiveness is minimized, breast cancer surgery leads to a significant decrease in upper limb functional level and can bring about aforementioned movement system disorders [17,24]. 

There are very few scientific papers touching on the topic of motor control assessment and uncontrolled movement in patients after breast cancer surgery. Fisher and Insana state that scapula kinematics disorders are present in patients after breast cancer surgery. The authors insist that dynamic scapula assessment is necessary for a complete evaluation [25]. Borstad and Szucs also point out the possibility of scapulohumeral rhythm disturbances as a consequence of breast cancer surgical treatment. The phenomenon is related to a loss of global range of shoulder elevation. The authors denote that kinematic assessment plays an important role in the overall patient examination [26]. Our previous research also indicates that the motor control assessment is useful in this group of patients. The motor control analysis together with the identification of uncontrolled movement plays an integral role in the diagnostic process in modern physical therapy, in general. The incidence of uncontrolled movement in the shoulder girdle area can lead to tissue overload and in turn cause other symptoms in the upper quadrant. The uncontrolled movement may be responsible for microtrauma summation over time, which can contribute to the development of new pain syndromes and/or further sensitization in already sensitized areas [10]. For these reasons, uncontrolled movement should be considered as a possible pain generating factor among breast cancer patients treated surgically. The complaints of pain in the study group included the area of the upper quadrant of the body in the region of the shoulder complex. Within the study group, the complaints were chronic in nature. The nature of the complaints indicated the presence of a mechanical component of pain. The occurrence of pain or its intensity was related to the movement of the free part of the upper limb. Therefore, motor errors can significantly contribute to the presence of pain. The use of mobility assessment and motor control tests in the study made it possible to observe a marked loss of function. The localized abnormalities in the examined area give a strong presumption that such a situation predisposes to the formation of microtrauma. Looking globally at the function, it is the deviations in the distribution of movement between the joints of the upper limb and the changes in neuromuscular coordination that neuromuscular coordination was observed in the research group. It is known that the accumulation of loads over time can lead to greater overloading and this, in turn, can lead to sensitisation in the hyper-mobile areas and a predisposition to increased soreness. The present study showed the occurrence of such deviations in comparison to a control group irrespective of age. These changes were called motor control deficits. On the other hand, the DASH questionnaire results indicate the possible existence of a neurogenic component to the pain. This is indicated by peripheral symptoms at the level of the free part of upper limb. The character of the pain is mixed, on the one hand, mechanical factors cause a fluctuation of symptoms as well as peripheral symptoms indicate a possible irritation at the level of nervous structures. The proximity of the nervous structures of the brachial plexus in relation to the elements of the musculoskeletal system may predisposes this area to a mixed character of pain.

The outcomes of our research indicate that movement faults, i.e., motor dissociation impairments are significantly more prevalent among subjects treated surgically. Movement faults of this sort were diagnosed when the tests outcome was “xx”. Statistically significant outcomes were present in following tests: Arm abduction test in frontal plane (Table 1), arm abduction test in scapular plane (Table 1), and kinetic medial rotation test of the shoulder (Table 1). The analysis confirms that motor control is impaired in the tested group when compared with the control group. Within the tested group, the operated side had more double “x” outcomes in comparison with the unaffected side. The difference was statistically significant. It was observable in the following motor control tests: Arm abduction test in frontal plane (Table 2), arm abduction test in scapular plane (Table 2), and kinetic medial rotation test of the shoulder (Table 2). The results of the analysis denote that surgical intervention in the past affects motor control in aforementioned movements negatively. This motor control deficiency is related to discoordination between scapulothoracic and humeroscapular muscles. The loss of neutral scapula position during motion results from coordination deficits in serratus anterior and different parts of the trapezius muscle. 

The results clearly show that the operation side is likely to have its motor control impaired. What is also important is that all of the subjects from the tested group had experienced pain in the last 3 months before the evaluation and that the pain had not been present before the surgical treatment. Neuromuscular disorders after the surgical treatment probably result from multiple factors overlap. Among these factors, there are pain leading to movement patterns alterations and neuromuscular coping mechanisms; myofascial restrictions; fear of movement (kinesiophobia); and limb involvement in everyday activities. The DASH questionnaire results analysis indicates clearly that there was a higher degree of subjectively perceived disability of upper limb in the tested group. This outcome points to the surgery as a negative factor contributing to a number of disorders limiting the functionality of the upper limb across an extended period of time after the treatment. 

Other studies investigating the clinical presentation of subacromial impingement syndrome show that motor control disturbance leads to discoordination of the trapezius muscle, which is responsible for stabilizing the scapula [27,28]. The studies confirm that the presence of pain in the shoulder is related to stabilizer muscles dysfunction and movement pattern impairments [29,30]. The systematic evaluation of motor control impairments using dissociation tests should be an integral part of the clinical examination in physical therapy for breast cancer patients after the surgical treatment. A physical therapist should focus not only on passive and active mobility assessment, palpation for tactile sensitivity and elements of neurological examination, but also on motor control assessment using dissociation tests as an equally valid diagnostic tool [31]. The presence of uncontrolled movement can in itself cause pain and be a contributing factor in the development of musculoskeletal disorders such as subacromial impingement syndrome [10]. In breast cancer patients after surgery, impaired motor control may contribute to tissue sensitization in the shoulder region resulting from chronic tissue overload. Chronic tissue overload in turn relates to the presence of movement faults. There seems to be a positive feedback loop between motor control impairments and pain syndromes in this group of patients, whereas pain disrupts the neuromuscular patterns, and movement faults lead to tissue overload resulting in further damage and/or sensitization. Motor control impairments may also play an important role in chronic pain generation among these patients, even many years after the treatment, significantly decreasing the functional level of upper limb and reducing the quality of life. The outcomes of our previous research on breast cancer patients after surgery also suggest that there is a need to introduce the motor control evaluation into the diagnostic process. In our pilot study, the presence of motor control disorders in the shoulder girdle was shown on the operated side in patients qualified for latissimus dorsi breast reconstruction procedure [32]. For these reasons, motor control evaluation procedures should be developed and implemented among breast cancer surgery patients. 

This paper is a preliminary report constituting a starting point for further scientific inquiry on this subject. The results of our investigation indicate that motor control disorders in the form of motor dissociation impairments are statistically more prevalent among women after breast cancer surgery in comparison with the control group. The results of the intra-group analysis indicate that the surgical intervention side is more likely to develop motor control deficits than the nonsurgical side. On the one hand, it indicates an obvious negative influence of the past treatment. On the other hand, it allows identifying specific disorders at the level of the shoulder complex motor control. Historically, the direction of specific motor control deficits has neither been identified nor taken into consideration when formulating clinical hypotheses in this group of patients. It is very important to accurately identify the specific directions of movement that are not controlled during the evaluation process. This will form the basis for programming motor re-training therapy. Additionally, this study fills a gap in the available research related to upper limb dysfunction in women after breast cancer surgery since it demonstrates dissociative motor control assessment possibilities in the shoulder complex. It is a starting point for further investigation of the presented subject, so that it will be possible to improve the physiotherapeutic evaluation of such patients in the future in order to improve their quality of life.

## 5. Conclusions

Surgical interventions in the breast cancer treatment and other medical procedures affect the level of motor control negatively in this group of patients. Movement faults are statistically more prevalent in the tested group.

Movement faults are more prevalent on the operated side in the tested group. Motor control deficits include dissociation disorders observable in abduction tests and kinetic medial rotation tests of the shoulder in the tested group. 

Motor control evaluation in the form of dissociation tests should constitute an integral part of the diagnostic process in physical therapy in this group of patients.

The assessment of motor control using dissociation tests should therefore be one of the routinely performed diagnostic procedures in the physiotherapeutic evaluation of women after breast cancer surgery. Therefore, the existing impaired motor control should be taken into account as a potential factor contributing to the decline in upper limb function, alongside factors such as decreased passive mobility or the existence of scar tissue after oncological treatment.

There is an urgent need to develop generally accepted rehabilitation protocols for patients with breast cancer, taking into account a broad, modern, and comprehensive approach to physical therapy treatment.

## Figures and Tables

**Figure 1 healthcare-09-00973-f001:**
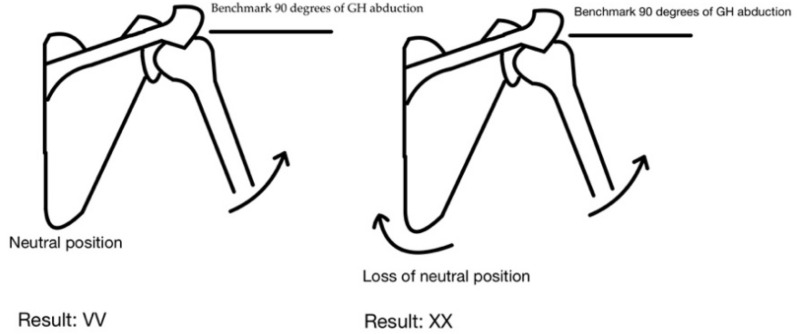
Dissociation test for the abduction direction—examples of outcomes.

**Table 1 healthcare-09-00973-t001:** Results of tests of glenohumeral (GH) control. Control and after the surgery group.

	Results (abduction control in scapular plane, right upper extremity)			
	xx	vx	vv	*p*-value (Fisher exact test for count data)
After surgery	8	3	10	0.02288
Control	1	5	16	
	Results (abduction control in scapular plane, left upper extremity)			
	xx	vx	vv	*p*-value (Fisher exact test for count data)
After surgery	9	7	5	0.0003468
Control	1	3	18	
	Results (abduction control in frontal plane, right upper extremity)			
	xx	vx	vv	*p*-value (Fisher exact test for Count Data)
After surgery	9	8	4	0.000184
Control	0	7	15	
	Results (abduction control in frontal plane, left upper extremity)			
	xx	vx	vv	*p*-value (Fisher exact test for Count Data)
After surgery	11	8	2	0.000006004
Control	0	7	15	
	Results (internal rotation control, right upper extremity)			
	xx	vx	vv	*p*-value (Fisher exact test for Count Data)
After surgery	10	8	3	0.02327
Control	3	9	10	
	Results (internal rotation control, left upper extremity)			
	xx	vx	vv	*p*-value (Fisher exact test for Count Data)
After surgery	5	10	1	0.004045
Control	2	8	12	

**Table 2 healthcare-09-00973-t002:** Results of tests of glenohumeral (GH) control. After surgery group, at the site of surgery.

	Results (abduction control in scapular plane, right upper extremity)			
	xx	vx	vv	*p*-value (Fisher exact test for Count Data)
Right	8	0	0	0.000004914
Left	0	3	10	
	Results (abduction control in scapular plane, left upper extremity)			
	xx	vx	vv	*p*-value (Fisher exact test for Count Data)
Right	1	4	5	0.004049
Left	8	4	0	
	Results (abduction control in frontal plane, right upper extremity)			
	xx	vx	vv	*p*-value (Fisher exact test for Count Data)
Right	8	0	0	0.00004914
Left	1	8	4	
	Results (abduction control in frontal plane, left upper extremity)			
	xx	vx	vv	*p*-value (Fisher exact test for Count Data)
Right	1	7	2	0.001395
Left	10	2	0	
	Results (internal rotation control, right upper extremity)			
	xx	vx	vv	*p*-value (Fisher exact test for Count Data)
Right	8	0	0	0.001307
Left	3	8	3	
	Results (internal rotation control, left upper extremity)			
	xx	vx	vv	*p*-value (Fisher exact test for Count Data)
Right	1	7	1	0.07939
Left	5	3	0	

## Data Availability

Not applicable.

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
