# Peer review of "Motor Control Evaluation as a Significant Component in Upper Limb Function Assessment in Female Breast Cancer Patients after Mastectomy"

_healthcare, 2021, doi:10.3390/healthcare9080973_

Round 1

Reviewer 1 Report

This is a very interesting topic, some comments could be considered. Both exclusion and inclusion criteria must ne justified and spelled out. The characters and type of pain at the shoulder must be described and defined. The validity and reliability of the DASH in the specific type, origin, age etc of patients/population must be justified and established. All methods including recording methods and software, motor control tests must be validated for the specific population and the investigation protocol must be standardized. Their chose over other available clinimetric tools must be justified. There is minimal critical analysis, comparison and contrast of the findings in this study with the ones in others.

Author Response

Dear Dr.

Thank you very much for your constructive comments, according to which we could improve our work. Below we will respond to specific comments and concerns that have been pointed out, thanks to which, in our opinion, the work has gained a lot.

Comments:

This is a very interesting topic, some comments could be considered. Both exclusion and inclusion criteria must ne justified and spelled out. The characters and type of pain at the shoulder must be described and defined. The validity and reliability of the DASH in the specific type, origin, age etc of patients/population must be justified and established. All methods including recording methods and software, motor control tests must be validated for the specific population and the investigation protocol must be standardized. Their chose over other available clinimetric tools must be justified. There is minimal critical analysis, comparison and contrast of the findings in this study with the ones in others.

Response:

Information on inclusion and exclusion criteria is described in detail in the manuscript.

Inclusion criteria: informed consent to participate in the study, age of majority, past surgical treatment for breast cancer treatment in the form of mastectomy or sparing treatment.

Exclusion criteria: lack of informed consent for participation in the study, for the study group – history of reconstructive procedure done with previous breast cancer surgery, less than two years recovery time after surgery, injuries in the form of fractures, dislocations in the shoulder girdle, shoulder pain before breast cancer surgery.

Information on pain complaints has been added in the discussion section.

The complaints of pain in the study group included the area of the upper quadrant of the body in the region of the shoulder complex. Within the study group, the complaints were chronic in nature. The nature of the complaints indicated the presence of a mechanical component of pain. The occurrence of pain or its intensity was related to the movement of the free part of upper limb. The use of mobility assessment and motor control tests in the study made it possible to observe a marked loss of function. The localized abnormalities in the examined area give a strong presumption that such a situation predisposes to the formation of microtrauma. Looking globally at the function, it is the deviations in the distribution of movement between the joints of the upper limb and the changes in neuromuscular coordination that neuromuscular coordination was observed in the research group. It is known that the accumulation of loads over time can lead to greater overloading and this, in turn, can lead to sensitisation in the hyper-mobile areas and a predisposition to increased soreness. The present study showed the occurrence of such deviations in comparison to a control group irrespective of age. These changes were called motor control deficits. On the other hand, the results of the DASH questionnaire indicate the possible existence of a neurogenic component to pain. This is indicated by peripheral symptoms at the level of the free part of the upper limb. The nature of the pain is mixed on the one hand mechanical factors cause fluctuation of symptoms and peripheral symptoms indicate possible irritation at the level of nervous structures. The proximity of the brachial plexus nerve structures to the elements of the musculoskeletal system may predispose this area to the onset of mixed pain.

The DASH questionnaire is used in the assessment of upper limb disability in a group of women after breast cancer surgery. It is a useful tool to assess the degree of disability of the upper limb based on questions relating to existing functional limitations in activities of daily living, the nature of complaints and the degree of limitation of function caused by their occurrence. This questionnaire is used in the clinical assessment of patients in many medical facilities, so its use promotes cooperation between healthcare centres. It has also been used in a number of scientific papers in breast cancer patients as a tool for assessing upper limb disability. The validity and reliability of the DASH questionnaire in our study group of women is confirmed by this article ( https://doi.org/10.1016/j.apmr.2013.07.022 ) in which the authors recommend the use of this questionnaire. This literature item has been added to the manuscript in the methods section (number 13).

In terms of the research method, motor control tests were used, the performance of which was recorded using digital cameras. As described during the presentation of the research method, the study protocol was standardised. This refers to the location of the cameras, the position of the patient and the order in which certain assessment procedures were performed. The study was conducted under standardised conditions; however, a detailed description in the manuscript would significantly lengthen the method section. The performance of the motor control tests and their evaluation was rigorously based on assumptions derived from the conception of Kinetic Control. The way the testing procedure is carried out, the determination of whether a given patient meets a given parameter and the interpretation of the results is described in detail in the book "Kinetic Control: The Management of Uncontrolled Movement" by Comerford M. Mottram S. and this was our primary source. The selection of motor control tests derived from the above mentioned physiotherapeutic method determined the necessity of referring to this type of disorders during physiotherapeutic assessment in this group of patients. It is an element of assessment used in our centre as a standard component of evaluation in the diagnostic and therapeutic process of patients in the described group in order to assess aspects of movement quality. The assessment of motor control is an area that is unfortunately neglected. The assessment of motor control itself is well established in the field of physiotherapeutic evaluation. Below are doi numbers to selected literature related to aspects of motor control including the utility of motor tests. Comerford M., Mottram S. "Movement and stability dysfunction - contemporary developments" (https://doi.org/10.1054/math.2000.0388), Mischiati C. Comerford M. "Intra and Inter-Rater Reliability of Screening for Movement Impairments: Movement Control Tests from The Foundation Matrix" (https://www.ncbi.nlm.nih.gov/pmc/articles/PMC4424474/) and the book "Kinetic Control: The Management of Uncontrolled Movement" by Comerford M. Mottram S. which we have already mentioned. Evaluation in terms of lymphedema, assessment of joint mobility of the shoulder complex or soft tissue flexibility in this group of patients are widely used elements. However, the qualitative aspects of movement are often neglected in clinical practice. However, one should be aware that this way of assessment is an innovative element in the field of physiotherapeutic assessment, which is not yet widely used. This state of affairs, in our opinion, is detrimental to patients as neglecting this aspect of assessment prevents the creation of a holistic picture of the patient's problem. The control group we studied allowed us to show statistically significant differences in the level of motor control compared to the study group. We are aware that the groups are not very large but the pandemic caused a large number of patients to drop out of the study and the study had to be suspended for a period. This study is a pilot. We are aware of the need to validate the tests used in the future. However, already at this stage we want to point out the occurrence of motor control disorders in the group of women after breast cancer surgery so that the qualitative assessment of movement is not neglected.

It is not possible to compare our results with results from other publications because there are no works yet relating to motor control disorders and their evaluation in this group of patients. As mentioned above, motor control assessment is an innovative element of physiotherapeutic evaluation used in our centre when working with patients in this group. Our work is pioneering, we want to draw attention to the problem of the occurrence of motor control disorders so that they are noticed during the evaluation process of patients in this group. We are also aware that this diagnostic-therapeutic area in the described group of patients must be developed. We plan to continue our research to develop this area.

Reviewer 2 Report

Thank you for the opportunity to review this manuscript. In this study, the authors attempt to identify specific uncontrolled movements in the area of the shoulder complex in a group of patients after surgical treatment of breast cancer. This reviewer believes that the following issues need to be addressed before recommending this manuscript for publication:

1- Including a video about how the measurements were performed in patients would clarify methods significantly and allow reproducibility.

2- Was the difference in DASH score significant between comparison groups?

3- Did you perform a power analysis for DASH score comparison?

Thanks again for the opportunity to review this manuscript!

Author Response

Dear Dr.

Thank you very much for your constructive comments, according to which we could improve our work. Below we will respond to specific comments and concerns that have been pointed out, thanks to which, in our opinion, the work has gained a lot.

Comments:

Thank you for the opportunity to review this manuscript. In this study, the authors attempt to identify specific uncontrolled movements in the area of the shoulder complex in a group of patients after surgical treatment of breast cancer. This reviewer believes that the following issues need to be addressed before recommending this manuscript for publication:

 1- Including a video about how the measurements were performed in patients would clarify methods significantly and allow reproducibility.

 2- Was the difference in DASH score significant between comparison groups?

 3- Did you perform a power analysis for DASH score comparison?

Response:

1- The patients only consented to the recording of digital footage for the study. Of course we have video footage of the study. Unfortunately, the subjects did not agree to the publication of their images. This state of affairs must be understood, as the examined women were often mutilated in the course of breast cancer treatment. Participation in the study itself was a very stressful experience for them. As far as the procedure of performing the tests is concerned, it is described in detail in the book "Kinetic Control: The Management of Uncontrolled Movement" by Comerford M. Mottram S. In the manuscript we have repeatedly referred to this book.

2- The mean score of the DASH questionnaire in the study group was 37.07 and in the control group it was 0.68. These values were calculated based on the formula found in the questionnaire manual. A higher score indicates a higher level of disability.

3- p-value=0, method Wilcoxon-Mann-Whitney Test

Reviewer 3 Report

The authors present their work on motor control evaluation in upper limb function assessment  in female breast cancer patients after mastectomy.

Even though their work is well structured and presented, I would suggest them to improve the presentation of their results in the manuscript section.

Author Response

Dear Dr.

Thank you very much for your constructive comments, according to which we could improve our work. Below we will respond to specific comments and concerns that have been pointed out, thanks to which, in our opinion, the work has gained a lot.

Comments:

The authors present their work on motor control evaluation in upper limb function assessment in female breast cancer patients after mastectomy.

Even though their work is well structured and presented, I would suggest them to improve the presentation of their results in the manuscript section.

Response:

This way of presenting the results was specified during the review in the International Journal of Environmental Research and Public Health so that the data were organised in two tables and additional elements were included as supplementary material.

Reviewer 4 Report

Extensive syntax and grammar editing should be accomplished. I suggest a careful revision of the entire manuscript to improve readability.

In my opinion, the introduction is a bit wordy. It should be shortened and focused.

ALND was performed in tested group?

The "results" section should be extensively revised. In my opinion, the results are not clearly presented.

Could the different mean age be a confounder? (tested vs control group)

Conclusion section is way to long. I suggest focusing this section on the real hearth of the present study. 

Moreover, if the authors decide to move some paragraphs of the conclusion to "discussion" section, I also suggest to revise the discussion and shorten it by 20%.

Author Response

Dear Dr.

Thank you very much for your constructive comments, according to which we could improve our work. Below we will respond to specific comments and concerns that have been pointed out, thanks to which, in our opinion, the work has gained a lot.

Comments:

1-In my opinion, the introduction is a bit wordy. It should be shortened and focused.

2-ALND was performed in tested group?

3-The "results" section should be extensively revised. In my opinion, the results are not clearly presented.

4-Could the different mean age be a confounder? (tested vs control group)

5-Conclusion section is way to long. I suggest focusing this section on the real hearth of the present study. 

6-Moreover, if the authors decide to move some paragraphs of the conclusion to "discussion" section, I also suggest to revise the discussion and shorten it by 20%.

Response:

1- Thank you for your suggestion. The introduction has been shortened.

2- Yes

3- This way of presenting the results was specified during the review in the International Journal of Environmental Research and Public Health so that the data were organised in two tables and additional elements were included as supplementary material.

4- Regarding the age of the patients: patients in the tested group that volunteered to participate in the study happen to be older on average than the volunteers from the control group. The difference in the mean age value is partly due to the participation of young women in the control group – women in the second and third decade of life. This is the reason for lower age mean value. However, the control group also included women in their sixth and seventh decades of life. Analysis of the results in control group demonstrates that dissociative movement disorders in the described directions are not as frequent as in the tested group. Another important aspect is the fact that some of the patients in the study group were not assessed in individual tests due to the existence of passive movement limitations preventing the test procedure. Regarding the influence of age on motor skills, it is assumed that the age of the patient does not negatively affect this parameter. The control group, despite its lower age, represents the reference and standard of fitness for the described tests, age is irrelevant. As mentioned above, in the control group, women in their sixth and seventh decade of life obtained test results indicating less frequent occurrence of movement dissociation disorders and, consequently, a higher level of motor skills when compared to the tested group. This also emphasizes that age does not have a negative effect on motor control parameters.

5 and 6 - Both sections have been rebuilt as instructed.

Round 2

Reviewer 4 Report

The following comments, from the first round of review where not properly addressed within text.

-In my opinion, the introduction is a bit wordy. It should be shortened and focused.

-ALND was performed in tested group? 

- I also suggest to revise the discussion and shorten it by 20%.

Author Response

Dear Dr.

Thank you very much for your constructive comments, according to which we could improve our work. Below we will respond to specific comments and concerns that have been pointed out, thanks to which, in our opinion, the work has gained a lot.

Comments:

In my opinion, the introduction is a bit wordy. It should be shortened and focused.

Response:

The introduction section has been shortened and focused.

Comments:

ALND was performed in tested group? 

Response:

Information was added in the description of the study group.

Comments:

I also suggest to revise the discussion and shorten it by 20%.

Response:

The discussion was revised and shortened.